# Pineapples' Detection and Segmentation Based on Faster and Mask R-CNN in UAV Imagery

**Yi-Shiang Shiu** [1] 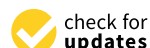**, Re-Yang Lee** [2,*] **and Yen-Ching Chang** [3]

1    Department of Urban Planning and Spatial Information, Feng Chia University, Taichung 407802, Taiwan
2    Department of Land Management, Feng Chia University, Taichung 407802, Taiwan
3    Chung Hsing Surveying Co., Ltd., Taichung 403006, Taiwan
\*    Correspondence: rylee@fcu.edu.tw

**Abstract:** Early production warnings are usually labor-intensive, even with remote sensing techniques in highly intensive but fragmented growing areas with various phenological stages. This study used high-resolution unmanned aerial vehicle (UAV) images with a ground sampling distance (GSD) of 3 cm to detect the plant body of pineapples. The detection targets were mature fruits mainly covered with two kinds of sun protection materials—round plastic covers and nets—which could be used to predict the yield in the next two to three months. For round plastic covers (hereafter referred to as wearing a hat), the Faster R-CNN was used to locate and count the number of mature fruits based on input image tiles with a size of $256 \times 256$ pixels. In the case of intersection-over-union (IoU) > 0.5, the F1-score of the hat wearer detection results was 0.849, the average precision (AP) was 0.739, the precision was 0.990, and the recall was 0.743. We used the Mask R-CNN model for other mature fruits to delineate the fields covered with nets based on input image tiles with a size of $2000 \times 2000$ pixels and a mean IoU (mIoU) of 0.613. Zonal statistics summed up the area with the number of fields wearing a hat and covered with nets. Then, the thresholding procedure was used to solve the potential issue of farmers' harvesting in different batches. In pineapple cultivation fields, the zonal results revealed that the overall classification accuracy is 97.46%, and the kappa coefficient is 0.908. The results were expected to demonstrate the critical factors of yield estimation and provide researchers and agricultural administration with similar applications to give early warnings regarding production and adjustments to marketing.

**Keywords:** object detection; sematic segmentation; zonal statistics; yield estimation

## 1. Introduction

Pineapples are one of the most critical economic fruit crops for many tropical areas. The top three countries in the world for pineapple production are the Philippines, China, and Costa Rica [1]. However, the weather quickly affects the quality and yield of pineapples, resulting in price and trade revenue fluctuation in many countries. With the development of agricultural technology, farmers can adjust the harvest period of crops following their needs and the weather conditions; this adjustment also increases the difficulties related to harvest time and yield survey for competent authorities. The primary pineapple yield survey method for the authorities is to request field investigators to conduct sampling visits to obtain farmers' planting areas and estimated yield. If the weather conditions permit, surveys can be conducted by remote sensing techniques such as equipping sensors on satellites and manned or unmanned aerial vehicles [2,3]. However, even with remote sensing techniques, the image interpretation task may also be labor-intensive. Especially in highly intensive but fragmented growing areas with various phenological stages, the image interpretation of fruit development close to harvest time needs high-spatial-resolution images and professionally trained experts.

In recent years, based on the development of machine learning, object detection and segmentation have also become two major research topics. Detection and segmentation can

be achieved by extracting image features containing shape, texture, and spectral information from multispectral bands [4–7]. Deep learning is a branch of machine learning developed based on several processing layers containing linear and nonlinear transformations, and it may be used as a reference for monitoring the cultivation area of pineapples as well as other crops in different growing seasons along with yield estimation [8,9]. Deep learning is a branch of machine learning whose algorithms are artificial neural networks (ANN) that mimic the functions of the human brain [10]. DNN can learn and summarize the characteristics of data in a statistical way based on myriad data with the opportunity to surpass the accuracy of human interpretation [11]. Convolutional neural networks (CNN), on the other hand, add procedures such as convolution, pooling, and flattening before the input layer and then enter the fully connected layers that are common in general ANN to construct prediction models [12].

There are many applications of CNN in remote sensing image classification, ranging from spatial resolution beyond tens of meters to centimeter-level aerial or UAV imagery. CNN can process and interpret remote sensing imagery including single-band panchromatic imagery, multispectral imagery above three bands, or hyperspectral imagery with hundreds of bands. For example, if the spatial and temporal adaptive reflectance fusion model (STARFM) is used to fuse Landsat 8 imagery with a spatial resolution of 30 m and MODIS imagery with a spatial resolution of 250–500 m, we can simulate daily imagery with spatial resolution of 30 m. In doing so, STARFM could generate continuous phenological parameters such as normalized difference vegetation index (NDVI) and land surface temperature (LST). For paddy rice, which has a relatively homogeneous spatial pattern, patch-based deep learning ConvNet could be applied to extract information helpful for distinguishing paddy rice and other crops [13]. Concerning hyperspectral imagery, the large number of bands might burden the hardware; thus, a previous study was initiated to develop minibatch graph convolutional networks (GCN) and further train large-scale GCN via small batches. Minibatch GCN would be able to deduce out-of-sample data performance without retraining the network and improving classification. Furthermore, GCN and CNN could be combined to extract different hyperspectral features to obtain key information for distinguishing different ground features in urban areas [14]. Regarding aerial photographs, which have higher spatial resolution by being provided with detailed information of texture and spatial correlation, CNN could identify enough features, even if there are limited visible/near-infrared bands. Thus, the previous study added the results of principal component analysis in aerial photographs to classify 14 types of forest vegetation with VGG19, ResNet50 and SegNet [15]. If very-high-resolution imagery (<0.5 mm/pixel) is available, pixel-level detection such as *Alternaria solani* lesion-counting in potato fields can be achieved with the U-Net model [16].

Depending on the purpose, CNN can process images through two fundamental perception methods: object detection and semantic segmentation [17]. The target detection method of deep learning is widely used, including malaria target detection [18], face detection [19], vehicle detection [20], and marine target detection [21]. In the "You Only Look Once" (YOLO) series, the Regions-Convolutional Neural Network (R-CNN), Fast R-CNN, and Faster R-CNN are widely known. The advantage of the YOLO series is that its recognition speed is fast and can meet immediate requirements. The application examples include using the YOLO version 3 (YOLOv3) model with four scale detection layers (FDL) [22], YOLOv5 to detect internal defects in asphalt pavements with 3D ground-penetrating radar (GPR) images [23], or YOLOv7 to detect and classify road surface damage with Google Street View images [24]. In contrast, the speed of the R-CNN series is slower, and the advantage is that the commission and omission recognition error rate may be lower [25]. Several previous studies have proposed related applications in fruit tree and fruit detection—for example, the R-CNN improved from the AlexNet network that was applied to detect the branches of apple trees in pseudo-color and depth images. In this state, the average recall and accuracy can reach 92% and 86%, respectively, when the R-CNN confidence level of the pseudo-color image is 50% [26]. MangoYOLO, improved and

developed from YOLOv3 and YOLOv2 (tiny), was designed to achieve fast and accurate detection of mango fruit in images of tree canopies, and the F1-score also reached above 0.89 [27]. With the advantages of fast and real-time detection, MangoYOLO has also been revised and used in real-time videos for fruit-detection applications [28]. With regard to Faster R-CNN, it has also been used to detect four different states of apple fruit with an average precision of 0.879 [29].

In detecting other fruits, such as mature and semi-ripe tomato counting, the previous study collected images and synthetic images as training samples. It then proposed an improved method of Inception-ResNet, which effectively improved the counting speed under the influence of shadows [30]. According to the different kiwifruit states (including occluded fruit, overlapping fruit, adjacent fruit, and separated fruit), it was concluded that the recognition rate of kiwifruit is 92.4% with Faster R-CNN [31]. Some potential applications of this technology include improving automated harvesting technology and using it for non-performing rate detection or pest inspection. For example, agricultural pest detection was proposed to reduce blind agricultural drug use to compensate for the lack of agricultural workers' knowledge of pests and diseases [32]. However, the aforementioned agricultural application documents are mostly images taken on the ground, and a small proportion of them were used for aerial photos.

Deep learning usually requires many samples to be effective [33]. For this, images are sometimes cropped into hundreds of smaller samples, and the target has to be manually labeled. A similar way of working can be seen in the study of pineapple detection with UAV images related to this study [2]. However, even with deep learning models such as Keras-RetinaNet, it is still necessary to shoot high-resolution UAV images at a low altitude of about 70 m to calculate the number of pineapple plants. The anchor box calculation of object detection includes the probability of whether or not it is a target. The detection result may be more likely to appear in outside areas such as fruit leaves, branches, and even other plants with higher heights, which interferes with the identification performance and computing time [2]. The situation of occlusion interference may be more difficult in aerial photos, and so it is necessary to improve it in combination with other methods, such as semantic segmentation, which is one of the options.

Semantic segmentation can improve the problem when object detection is disturbed by irrelevant objects. It can be achieved by being labeled in pixel units and obtaining category labels for different parts of objects in one image [34]. For example, DeepLab and region growing refinement (RGR) based on CNN have been used to detect flowers of various kinds of fruits with different lighting conditions, background components, and image resolution [35]. Another application was mango detection by developing MangoNet based on CNN and architecture involving fully convolutional networks (FCN), which can also be used in different scales, shading, distance, lighting, and other conditions [36]. The Mask R-CNN combines the two-stage model of Faster R-CNN, and the feature pyramid network (FPN) method was included to make predictions using feature maps with high feature levels in different dimensions. Mask R-CNN also improved the shortcomings of a region of interest (ROI) pooling in Faster R-CNN and caused the precision of the bounding box and object positioning to reach the pixel level. For the improvement of the accuracy of object boundary description, Mask R-, which includes the concept of FCN and CNN, can achieve perfect instant segmentation of objects [37]. In a previous study on potato plants and lettuces for single plant segmentation, Mask R-CNN achieved a mean average precision (mAP) of 0.418 for potato plants and 0.660 for lettuce. In the detection, the multiple object tracking accuracy (MOTA) of potato plants and lettuces can also reach 0.781 and 0.918, respectively [38].

From the literature analysis presented above, it can be seen that semantic segmentation can classify pixels in images, identify the categories and positions in photos, and improve the problem of object detection being impeded by irrelevant objects. Therefore, the objectives of this study were to: (1) use object detection to detect pineapple fruit; (2) use semantic segmentation to distinguish the distribution of pineapples and non-pineapples;

(3) combine the results of detection and segmentation to gain detailed yield estimation. Based on the experience of previous studies [2,3], we used high-resolution unmanned aerial vehicle (UAV) images with a ground sampling distance (GSD) at the centimeter level to detect the plant body of pineapples. The results are expected to grasp the critical factors of yield estimation and provide researchers and agricultural administration with similar applications to give early warnings regarding production and adjustments to marketing.

## 2. Materials and Methods

### 2.1. Study Area

The study area Chiayi County in Central Taiwan is representative of the tropical and subtropical climate pineapple production area. In FAOSTAT 2020, Taiwan ranked 8th in yield at 53,196 kg/ha, 18th in production at 419,028 tons, and 28th in harvest area at 7877 ha [1]. According to the Agricultural Situation Report Resource Network in 2020 [39], Chiayi County has the highest yield in Taiwan, reaching 56,130 kg/ha.

Chiayi County is the transition zone between the tropical and subtropical monsoon climates [40]. The main producing area of pineapples is the plain area, and the monthly rainfall exceeds 200 mm, mainly between June and September. Months with a rainfall below 50 mm are concentrated from October to February, and the month with the lowest rainfall is November, at only about 21 mm. The mean annual precipitation was 1821.6 mm during the period of 1991 to 2020. In terms of monthly average temperature, the annual average monthly temperature is above 15 °C, and monthly average temperatures above 20 °C reach up to 9 months, indicating a long summer and harsh winter climate. Regarding the monthly average temperature changes, the highest is 28.9 °C, and the lowest is 16.8 °C, so it is suitable for the growth of pineapples.

Based on the pineapple yield per hectare announced by the Agricultural Situation Report Resource Network in 2020, and the prohibited and restricted navigation range announced by the Civil Aviation Administration in Taiwan, we overlaid the main agricultural areas with pineapple on the 372.060 ha selected study area that allowed aerial photography (Figure 1). The total planting area of pineapples in the study area is 152.387 hectares. The base image is the UAV image with GDS 3 cm and red, green and blue bands (RGB image), taken on 4 April 2021.

The main pineapple species in the study area is *Ananas comosus*. In different phenological growth stages, the growth patterns can be divided into the plant, growing, forcing, open heart, flowering, mature, harvest, and sucker stages [41]. Figure 2 presents the growth period of pineapples in this study area, which takes about 18 months. In general, each field could be harvested once every two years or even twice every three years with similar frequency. The pineapple fruit is primarily available from March through July and other sporadic months. Considering the situation, farmers may adjust the production period by batching, forcing flowers to maintain the continuous production of pineapples in the field. The batching process can also avoid problems such as lack of work or market price fluctuations.

After interviews with local farmers, it was found that the "forcing stage" is the key to ripening pineapple fruit. The so-called forcing stage refers to artificially forcing flowers, mainly using calcium carbide and ethephon. More importantly, pineapples in the "mature" and "harvest stages" are more prone to sunburn after the forcing stage because the local sunshine hours are long and the intensity is high. There are various sun protection methods, such as covering with round plastic covers (the yellow plastic materials in a mature stage, shown in Figure 2) or covering with nets [42]. According to the descriptions from the local farmers, these kinds of protection are maintained for about two to three months. In this study, we call the fruits at this stage "sunscreened mature fruits." If farmers cover fruits with a round plastic cover, there is a bonus effect on the taste. Hence, farmers mostly choose this time-consuming and labor-intensive method based on quality management requirements. If farmers cover the fruit with a black net, it can reduce working time and

save labor; this makes the taste of the fruit slightly worse. Because covering with nets can reduce part of the work cost, this method of sun protection is also used in some fields.

If the number of plants per unit area of sunscreened mature fruits can be calculated, the yield in the next two to three months can be established. Therefore, this study takes the sunscreened mature fruit as the primary research object and combines object detection and semantic segmentation to estimate the current pineapple yield.

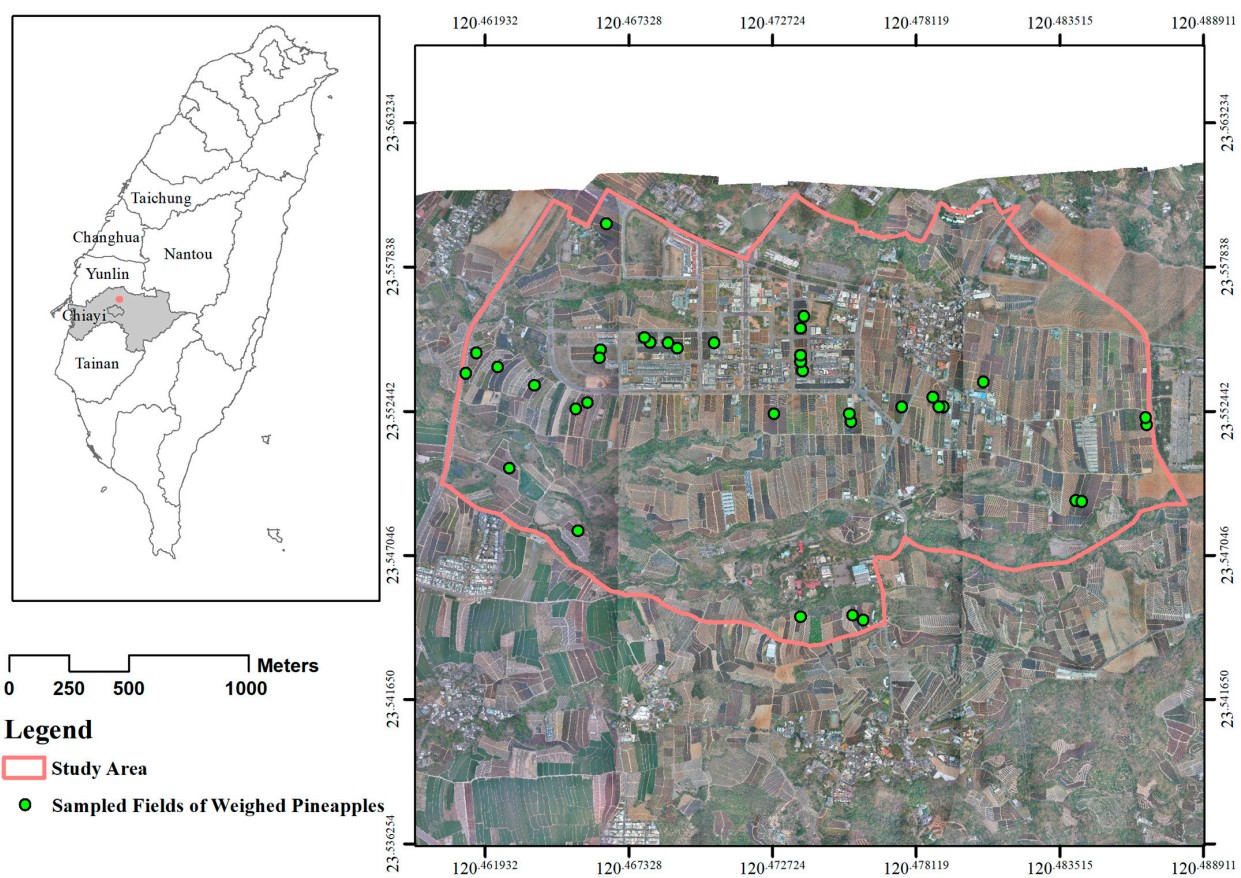

**Figure 1.** Study area, sampled fields and the UAV image with GSD 3 cm.

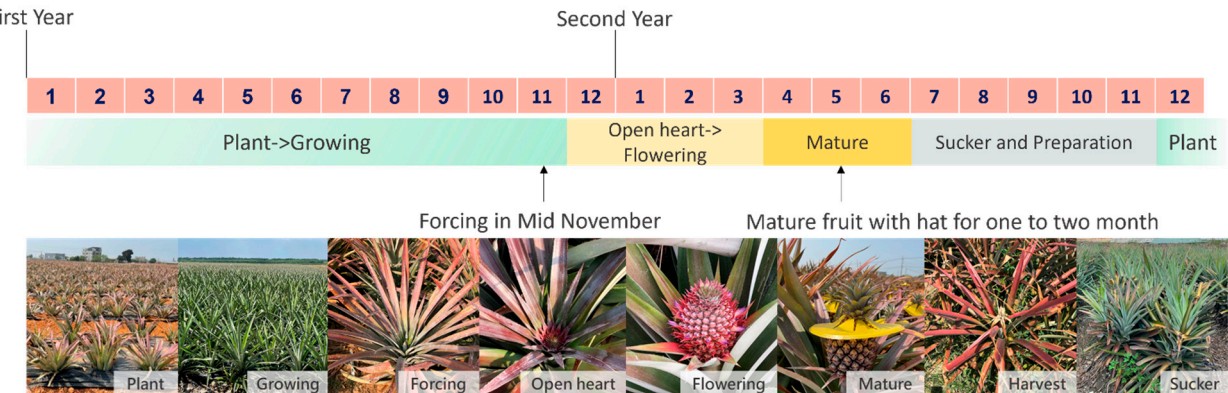

**Figure 2.** The phenological growth stages of pineapple in the study area.

### 2.2. Materials

With the aim of providing more practical feasibility for a larger scale in other areas worldwide, this study took 3 cm-spatial resolution UAV images as the test target, obtained on 4 April 2021. The camera model used was SONY ILCE-7RM2, and each photo was

recorded as an 8-bit RGB image with a pixel size of 7952 × 5304. The UAV was roughly 200 m above the ground when shooting in the air, and the coverage rate of each vertical shot exceeded more than 70%. The Pix4D Mapper image processing software was then used to solve the internal and external orientation parameters of the automatically matched feature points in the image to produce the digital terrain model for the final orthorectification processing, which can be used for subsequent target detection and semantic segmentation. The final GSD of the mosaicking image was better than 3 cm (inclusively).

During the shooting by the UAV, ground investigators were dispatched to the local area to record the growth status of pineapples and farmers' farming habits, which could facilitate the subsequent training and testing of sample construction for deep learning models. The investigators marked the pineapple planting location and phenological growth pattern of each cultivation field on the cadastral map based on Figure 2. Figure 3 shows the in situ condition of mature fruits covered with different materials, which were our main targets of yield estimation with UAV images.

The concepts above in Section 2.1 illustrates that the yield in the next two to three months can be grasped if the number of mature sunscreen fruits can be calculated. Therefore, in addition to investigating pineapple planting areas and growth patterns, we also weighed mature fruits by randomly selecting the sample cultivation fields. The number of weighed pineapples varied depending on the number of pineapples that farmers had not yet sorted, graded, and packaged. The weighed fruit number ranged from 50 to 70. Combined with official records, the total number of sampled cultivation fields was 36 (green dots in Figure 1). The average weight of pineapples in the sampling field was 1.507 kg, and the standard deviation was 0.246 kg.

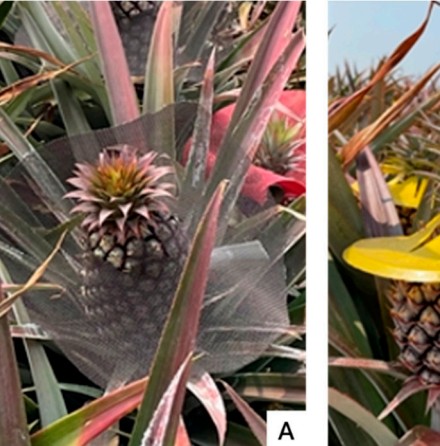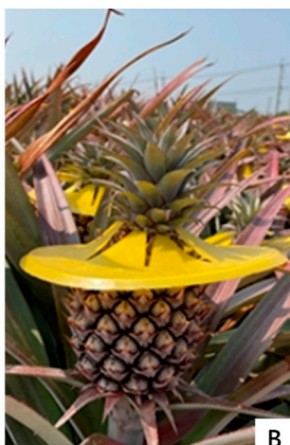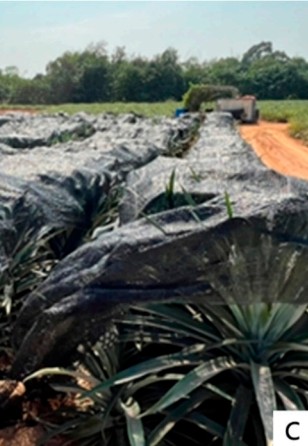

**Figure 3.** Photos of the three kinds of "sunscreen mature fruits" pineapples in the study area covered with: (**A**) plastic sheets of different colors; (**B**) yellow plastic covers; and (**C**) covered with nets.

### 2.3. Object Detection

As in Figure 3, there were three kinds of mature sunscreened fruits in our study area. To simplify the description, we refer to mature fruits covered with "plastic sheets of different colors" (Figure 3A) and "yellow plastic covers" (Figure 3B) as "hats." Because hats can more clearly identify each fruit, performing a more accurate yield estimation is possible. The characteristics of fruits "covered with nets" (Figure 3C) are different from the hats and will be discussed in the next Section 3. In this study, the object detection method, Faster R-CNN, was adopted to calculate the number of hats.

The model of Faster R-CNN can be divided into four parts (Figure 4):

(a) Convolutional layers;
(b) Region proposal networks (RPN);
(c) ROI pooling;
(d) A classifier, as described in the following Section 2.4.

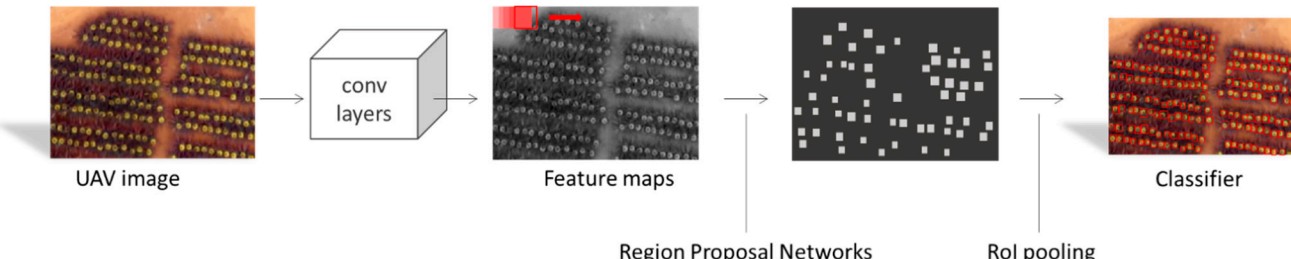

**Figure 4.** The framework of pineapple detection with Faster R-CNN.

Regarding convolutional layers, the convolution operation extracted features from the image, such as the object boundary and shape of the fruit on the image. The convolution kernel was used to inspect the local area and perform a dot product operation to obtain the eigenvalues of the area. Nevertheless, the convolution kernel did not operate with a whole image to prevent an excessively high amount of calculation. It is usually set on two parameters, kernel size and stride, for the convolution kernel to slide up, down, left, and right through an image. The extracted feature maps were processed in the subsequent RPN and fully connected layers, and ResNet50 was used as the backbone.

A moving window was generated on the last convolutional layer to generate bounding boxes similar to the hats. At each moving window position, multiple bounding boxes were predicted simultaneously, and the probability score of each target object was estimated. The center of the moving window was the anchor. A previous study observed the impact of anchors with different aspect ratios and scale combinations on performance [43]. We referred to it and used three aspect ratios and three scales to generate nine anchor boxes. We assigned each anchor a binary class label (hat or non-hat) for training the RPN. We assigned a positive label based on the hats: (1) the anchor overlapped the ground truth box with the highest intersection over union (IoU), or (2) the IoU overlap ratio of the anchor point and any ground-truth box was higher than 0.7. We removed all of the moving windows, including blank margin areas in the image, to prevent a high amount of calculation. Next, the non-max suppression (NMS) was used to filter all remaining windows, and the IoU did not exceed 0.7. The final goal was to control the anchor boxes to about 2000. These 2000 boxes were not calculated every epoch; instead, we randomly selected 256 positive samples (IoU > 0.7) and 256 negative samples (IoU < 0.3) as mini-batch data. A fully connected network was then inputted to obtain the probability of an object, the probability of not containing an object, the XY coordinate position, and the width and height of the prediction area as the proposed bounding box. Briefly, the purpose of RPN was to improve operational efficiency.

RoI pooling then collected bounding boxes proposed by RPN and calculated proposal feature maps to classifiers. With two fully connected networks, the class of each object was exported through the normalized exponential function (softmax) and the region of that with linear regression. These procedures made the bounding box more accurate, and finally, Faster R-CNN achieved hat detection with its coordinate position.

On the other hand, different hat sizes may have different effects when training the Faster R-CNN model. Compared with smaller objects, larger objects lead to more significant errors. To reduce these effects, normalization was used to improve the loss calculation of the width and height of the bounding box. The improved loss function is as follows:

$$L(\{p_i\}, \{t_i\}) = \frac{1}{N_{cls}} \sum_i L_{cls}(p_i, p_i^*) + \lambda \frac{1}{N_{reg}} \sum_i p_i^* L_{reg}(t_i, t_i^*), \tag{1}$$

where $i$ is the anchor of the mini batch, $p_i$ is the predicted probability of the anchor, $p_i^*$ is the real label (one for a positive anchor and zero for the negative one), $t_i$ is the coordinate representing the predicted bounding box, $t_i^*$ is the ground-truth box associated with the positive anchor, and $L_{cls}$ is the loss function for two categories (hat and non-hat). For

regression loss, $L_{reg}(t_i, t_i^*) = R(t_i - t_i^*)$, $R$ is the robust loss function, and $p_i^* L_{reg}$ indicates that the regression loss is only activated for positive anchors ($p_i^* = 1$). The outputs of *cls* and *reg* consist of $\{p_i\}$ and $\{t_i\}$, respectively. The normalized $N_{cls}$ and $N_{reg}$ are weighted by the balance parameter $\lambda$. By default, we set $\lambda = 10$, so the weights of *cls* and *reg* are roughly the same.

The UAV images used in this study were orthorectified during the preprocessing, as there are few available datasets with an orthographic view of pineapples with hats. Therefore, we established a training dataset according to the study area's spatial distribution and patterns of fruits with hats. In other words, we tried to disperse the samples as much as possible in different areas and select hats of various shades of color and materials. To label each pineapple in every training image, we applied the built-in Label Objects for Deep Learning tool in ArcGIS Pro 2.8 to label the bounding box of the hats in the image, and the corresponding PASCAL VOC annotation files were generated after the labeling. The training sample consisted of image tiles of a fixed size. The UAV images were cropped to $256 \times 256$ pixels with a stride of 128 for each tile. Each tile covered several fruits with hats. We set the training to verification sample ratio as 6:4, and the max epochs were 300.

When the IoU between the machine-predicted bounding box and the manually labeled bounding box was more significant than 0.5, the hat was considered correctly detected and regarded as a true positive. However, errors can occur if there are multiple predictions in the same local area or authentic multiple hat clusters in the same area. Each machine-predicted and human-labeled box is calculated only once to avoid this problem. If the machine-predicted box is not included in the human-labeled box, it is a false positive. Finally, after evaluating all predicted boxes and classifying them as true positives, false positives or false negatives, the remaining set of annotated boxes were considered true negatives.

The confusion matrix in Table 1 can calculate precision and recall. Precision is the proportion of the target's predicted target and shows whether the result is accurate when the model predicts the target. A recall is the ratio of the actual target to what is predicted to be the target and gives an idea of the model's ability to find the target.

$$Precision = \frac{TP}{TP + FP} \tag{2}$$

$$Recall = \frac{TP}{TP + FN} \tag{3}$$

**Table 1.** Performance assessment of Faster R-CNN in this study.

| | | Ground Truth | |
| --- | --- | --- | --- |
| | | **Hat** | **Non-Hat** |
| Faster R-CNN result | Hat | True Positive (TP) | False Positive (FP) |
| | Non-hat | False Negative (FN) | True Negative (TN) |

The F1-score is a commonly used indicator for evaluating the accuracy of a model. The score takes both false positives and missed calls into account. It is the harmonic mean of precision and recall. The maximum value is 1, and the minimum value is 0.

$$F1 - score = 2 \times \frac{Precision \times Recall}{Precision + Recall} \tag{4}$$

Since there is only one category of pineapple plant interpretation (whether hats are detected), the average precision (AP) evaluation model can be used, which is also a standard precision index. AP is the area under the precision–recall curve. Usually, we suppose that the precision maintains a high value with the recall increase. In this case, the model can be considered a good prediction model because the values of precision and recall are between

0 and 1, so AP is also between 0 and 1. We used $AP_{50}$ ($AP^{IoU = 0.5}$), $AP_{75}$ ($AP^{IoU = 0.75}$), $AP_{95}$ ($AP^{IoU=0.95}$) to access the accuracy.

$$AP = \int_0^1 p(r)dr \tag{5}$$

*2.4. Sematic Segmentation*

As in Figure 3C, the third kind of mature sunscreened fruit in our study area was covered with nets. Although detecting the fruit under nets is impossible, mapping the area covered by the nets still contributes to yield estimation in the next two to three months.

The detection model for fruits covered with nets was built based on Mask Regions with Convolutional Neural Networks (Mask R-CNN) [37]. Mask R-CNN combines object detection and instance segmentation. It extends the existing network structure in Faster R-CNN by adding mask branches and consequently can obtain pixel-level detection results. Mask R-CNN can not only give each target object its bounding box but also mark whether or not the pixel belongs to the object in the bounding box.

The convolutional layer used a kernel of a specified size and a stride to slide sequentially from top to bottom and left to right to obtain each local feature in the image as the input of the next layer. After the summation of each local area in the image was obtained, the linear rectification function (rectified linear unit, ReLU) was used to output the eigenvalue and then was provided to the next layer for use. After the ReLU function processing, a value less than 0 was output as 0, and a value greater than 0 was output directly. This resulted in the feature map, and each point on the feature map could be regarded as the feature of the area in the original image, which can be passed to the next pooling layer. The following is the convolutional layer formula:

$$x_j^l = f\left(\sum_k^m \left(x_k^{l-1} * W_{kj}^l\right) + b_j^l\right), \tag{6}$$

where * is the convolution operation, $x_k^{l-1}$ defines the $l-1$st layer as the kth feature map, $x_j^l$ defines the lth layer as the *j*th feature map, *m* is the number of input feature maps, $W_{kj}^l$ is the weight also known as the kernel or filter, and $b_j^l$ is the error. *f* is the nonlinear activation function where the ReLU function is used, and the following is the ReLU function formula:

$$f(x) = \max(0, x) \tag{7}$$

The feature maps generated by convolution were trained on RPN to generate boxes similar to region proposals. A moving window was generated on the last convolutional layer. The feature map predicted the boxes of multiple regional proposals at each moving window position and estimated each proposal's object or non-object probability score. These candidate objects were then subjected to the pooling layer step, which reduced the spatial dimensions and complexity. In this way, global features could be extracted along with local features to reduce the parameters required by subsequent layers, thereby increasing the efficiency of system operation. After the last convolution, the pooling layer generated a size vector representing the output. Mask R-CNN used RoIAlign (Region of Interests Align, RoIAlign) to replace the pixel bias problem in the original RoI pooling and sent these outputs to fully connected layers and FCN, respectively. Next, they performed classification and instance segmentation. The fixed-size feature map was obtained through RoIAlign and sent to the detection network with different thresholds. The fully connected layer analyzed the previous frame output target coordinate value, converted it into probability through the softmax function, and outputted the result following the highest probability value category. FCN slices the image into pixels and outputs a mask of objects so that boundaries can be delineated along field edges by an overlay network. FCN is commonly used rather than the traditional CNN. This is because classification is performed after obtaining feature vectors of fixed range with fully connected layers. However, FCN changes all layers into

convolutional layers so that the image's dimensions can be divided into smaller ones to achieve pixel-level classification [44]. The framework of Mask R-CNN in this study is shown in Figure 5. The backbone used was ResNet101.

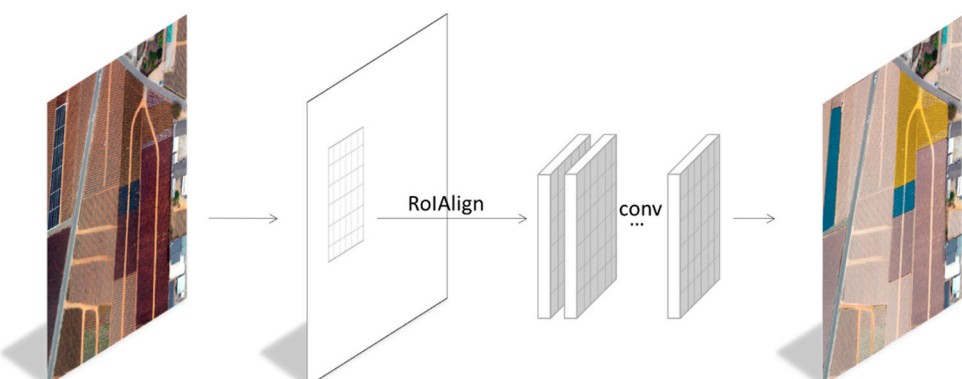

**Figure 5.** The framework of Mask R-CNN in this study.

Similar to Faster R-CNN, the UAV images used were orthorectified during the pre-processing, while there are few available datasets with such an orthographic view of fruits covered with nets. Therefore, we established a training dataset according to the study area's spatial distribution and patterns of fruits covered with nets. In other words, we tried to disperse the samples as much as possible in different areas and select black nets of various shades of color and materials. The irregular boundaries of nets for the training data were delineated by Labelme software according to the actual growth pattern of pineapples and stored in JSON format. The training sample consisted of image tiles of a fixed size. The UAV images were cropped to 2000 × 2000 pixels with a stride of 1000 for each tile. The 2000 × 2000 pixels were decided based on the study area's average width (about 20 m) of cultivation fields. Each tile with 2000 pixels, i.e., 60 m, can cover at least two cultivation fields covered with nets and other categories, which would be helpful for the machine to delineate the irregular field boundaries. We set the ratio of training to verification samples as 6:4, and the number of iterations was 300. The mean IoU (mIoU), $AP_{50}$, $AP_{75}$, and $AP_{95}$ [45,46] were used to evaluate the detection results for the nets. AP and IoU were defined in Section 2.3. The following are the formulae of the mIoU:

$$mIoU = mean\left(\frac{TP}{TP + FN + FP}\right) \qquad (8)$$

Finally, zonal statistics were used to sum up the area covered with nets based on each cadastral cultivation field to obtain the results. These results were also used to complement the Faster and Mask R-CNN results to improve the detection accuracy of various mature fruits. We also consider all land features including hat, net and background classes; the results for zonal statistics, user's accuracy, producer's accuracy, overall accuracy, and kappa coefficient were used for accuracy assessment [47]. The following are the formulae of the above indices:

$$user's\ accuracy = \frac{x_{ii}}{x_{i+}} \times 100\% \qquad (9)$$

$$producer's\ accuracy = \frac{x_{ii}}{x_{+i}} \times 100\% \qquad (10)$$

$$overall\ accuracy = \frac{\sum_{i=1}^{k+1} x_{ii}}{N} \times 100\% \qquad (11)$$

$$kappa\ coefficient = \frac{N\sum_{i=1}^{k+1} x_{ii} - \sum_{i=1}^{k+1} x_{i+} \cdot x_{+i}}{N^2 - \sum_{i=1}^{k+1} x_{i+} \cdot x_{+i}} \qquad (12)$$

where $k$ indicates the number of classes in the confusion matrix, i.e., two classes including net and hat, for a total of k + 1 classes (including a background class); $x_{ii}$ stands for the area correctly classified; $x_{ij}$ is the area predicted to be the background but is actually a positive label; $x_{ji}$ is the area predicted to be the foreground but is actually a negative label; $x_{i+}$ represents the total area classified as class $i$; $x_{+i}$ denotes the total area of class $i$ in ground truth data; $N$ indicates the total area of the study area.

## 2.5. Study Procedures

This study was divided into five parts to summarize the abovementioned materials and methods (Figure 6).

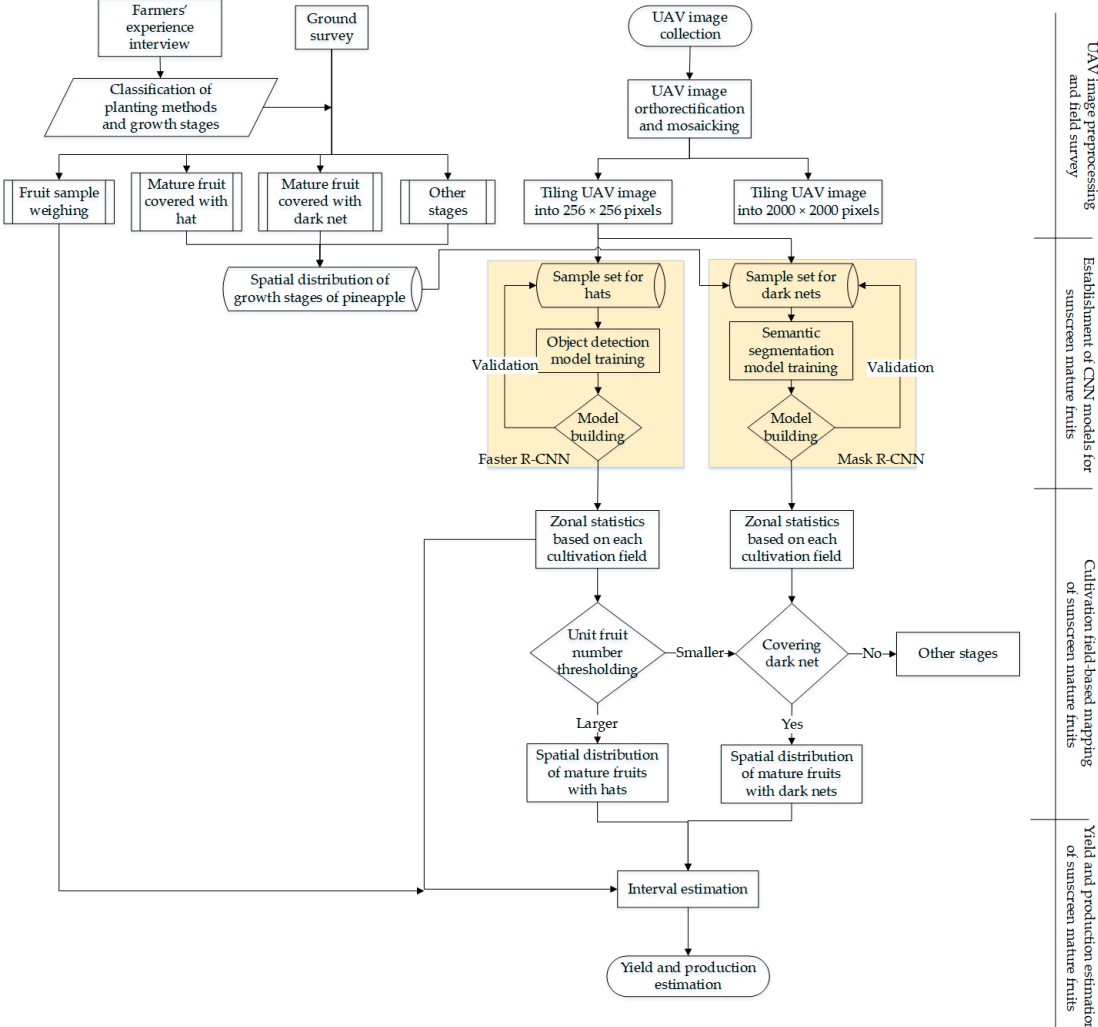

**Figure 6.** Flowchart of this study.

1. The first part involved the ground-truth data which were collected from the local area, obtaining the spatial distribution of various growth patterns of pineapples, and preliminarily establishing the growth stage of pineapples in the study area.
2. In the second part, we targeted hats (i.e., sunscreen mature fruits with plastic sheets of different colors and yellow plastic covers in Figure 3A,B) as the basis for estimating the yield per unit area. Because the shapes of hats are relatively consistent, the spatial distance and correlation of each hat are both consistent, respectively, and Faster R-CNN is a suitable tool for our purpose. We established corresponding training and

validating samples in a ratio of 6050:4010 plants (approximately 6:4) for these kinds of ripe pineapples.

3.  In the third part, we targeted nets (Figure 3C) to map the area of other sunscreen mature fruits. Dark nets have relatively consistent spectral and texture features, but most of them are irregular, so Mask R-CNN with semantic segmentation capability is an ideal model for our purpose. Similar to the Faster R-CNN model, we established corresponding training and validating samples at a ratio of 1.53:1.02 in ha (approximately 6:4) for ripe pineapples that covered the nets.

4.  In the fourth part, the fruit detection results were corrected according to the zonal statistics based on each cultivation field and thresholding. The zonal statistics operation calculates statistics on the number of hats and the area covered with nets within each cultivation field. The thresholding procedure was used to solve the potential issue of farmers' harvesting in different batches.

5.  The fifth part summarized the mature fruit planting area and estimated the study area's yield with interval estimation. Based on the detection results, the number of fruits per unit area was calculated, supplemented by the random sampling on-site and the official records of fruit weight in the study area.

The software and hardware environment of the proposed procedure was as follows: programming language, Python; auxiliary software, ArcGIS Pro 2.8; operating system, Windows 10; graphics card, NVIDIA® TITAN RTX GDDR6 24 GB; processor, Intel(R) Core(TM) i9-9900KF CPU @ 3.60GHz; and memory, DDR4 2666 MHz 64 GB.

## 3. Results

The results of this study were divided into three parts. The first involved using Faster R-CNN to detect hats and assess the F1-score, AP, precision, and recall accuracy. Second, Mask R-CNN was used to map net coverage. Since the Faster R-CNN model can only estimate the number of fruits wearing a hat, fruits under nets could not be detected, so Mask R-CNN was used to map other mature fruits. The results of Mask R-CNN can detect and correct some fields where omission and commission errors may occur in Faster R-CNN. Third, we finally used the detection results obtained in the first part to calculate the number of fruits wearing a hat per unit area. The fruit number per unit area was also adopted to estimate the yield of fields covered with nets. The weight data from the field survey and official statistics were used to estimate the pineapple yield in the study area.

### 3.1. Hat Wearing Detection

Although the mature fruits covered with nets could not be efficiently identified on the UAV image with Faster R-CNN, those with hats could be identified in this procedure. We randomly sampled the hat objects, as shown in Figure 7. Figure 7A shows the sample distribution of plastic sheets of different colors, and Figure 7B shows that of yellow plastic covers. The base image was a UAV image with GDS 3 cm used for the CNN model.

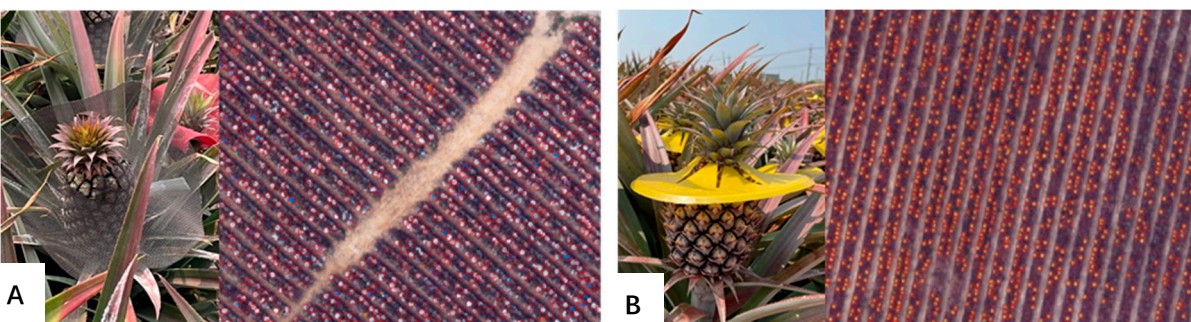

**Figure 7.** The in situ photos and sample set of selected plastic sheets of different colors (**A**) and yellow plastic covers (**B**).

The detection speed is 20.5 s per hectare and, in total, 2.16 h for the whole study area. The parameters of Faster R-CNN during the training process are listed in Table 2, and the accuracy assessment is shown in Table 3.

**Table 2.** Parameters of hat wearer detection with Faster R-CNN during the training process.

| Parameters | Base Learning Rate | Batch Size | Weight Decay | Momentum | Anchor Start Size | Aspect Ratios |
|---|---|---|---|---|---|---|
| Values | 0.001 | 2 | 0.0005 | 0.9 | 10 | [0.5, 1, 2] |

**Table 3.** Accuracy assessment of hat wearer detection results with Faster R-CNN.

| IoU | Precision | Recall | F1-Score | AP |
|---|---|---|---|---|
| 0.5 | 0.990 | 0.743 | 0.849 | 0.739 |
| 0.75 | 0.885 | 0.665 | 0.760 | 0.601 |
| 0.95 | 0.046 | 0.034 | 0.039 | 0.002 |

Regarding the recall value, although the hats could be accurately identified, the number of missed judgments was also high. The commission errors in most of the situations are shown in Figure 8, which contains the following:

6. Some land features which may be similar in color or shape to a large fruit wearing a hat.
7. A field area where hats and nets were used together.
8. A field area which is inconsistent in the maturity period and may lead to sporadic hats.

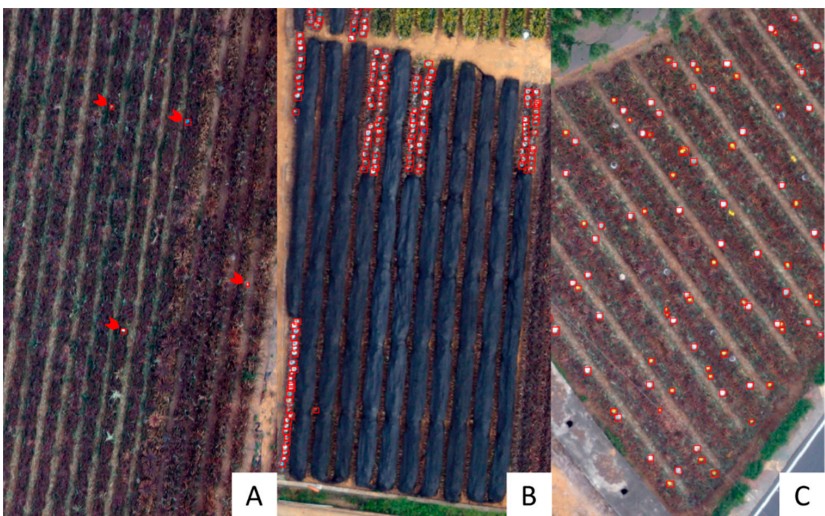

**Figure 8.** The commission error situations of hat wearer detection. (**A**) Commission error due to similarity of land features; (**B**) mixed use of wearing a hat and covered with nets; (**C**) inconsistency in field maturity, resulting in sporadic hats. The red boxes and red arrows indicated the hat detection results with Faster R-CNN.

To amend these commission errors, we referred to farmers' planting experience and the habit of harvesting. According to the Central Region Branch, Agriculture and Food Agency's official statistics, the maximal fruit number is 36,000 per hectare. Farmers are usually divided into five to ten harvests according to the size of the field. In other words, we may use a threshold proportion of mature fruit of 20% of 36,000 plants/ha (at least 20% of plants wearing a hat). Therefore, after summing up the detection number of fruits wearing a hat based on each field with zonal statistics, the fields with a number of hats per field of more than 7200 plants/ha were classified as a field with hats, whose area was

9.901 hectares, with a total of 370,960 plants. Those with less than 7200 plants/ha were marked and overlaid with the results of Mask R-CNN described later in Section 3.2.

### 3.2. Mapping of Area Covered with Nets

The photos of covering nets and UAV images are shown in Figure 9. In order to solve the problem of the detection model being unable to detect the pineapple fruits under nets, this section describes the use of Mask R-CNN to map the pineapple planting area covered with nets. If the area covered with nets can be mapped by Mask R-CNN, the number of plants per unit area obtained by Faster R-CNN and official statistics can be used to estimate the production of pineapples. The parameters of Mask R-CNN during the training process are listed in Table 4, and the detection results of the nets are listed in Table 5. The detection speed is 29.3 s per hectare and, in total, 3.08 h for the whole study area.

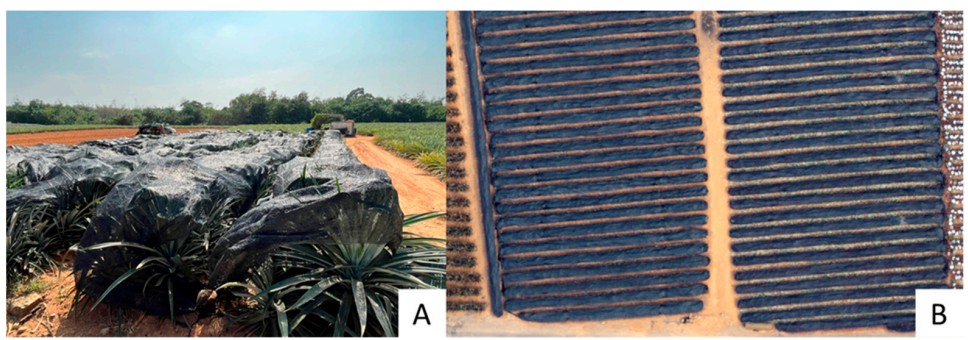

**Figure 9.** Photos of pineapples covered with nets from different perspectives: (**A**) in situ photos; (**B**) UAV image with GSD 3 cm.

**Table 4.** Parameters of net detection with Mask R-CNN during the training process.

| Parameters | Base Learning Rate | Batch Size | Weight Decay | Momentum | Anchor Start Size | Aspect Ratios |
|---|---|---|---|---|---|---|
| Values | 0.005 | 2 | 0.0001 | 0.9 | 32 | [0.5, 1, 2] |

**Table 5.** Accuracy assessment of net detection results with Mask R-CNN.

| $AP_{50}$ | $AP_{75}$ | $AP_{95}$ | mIoU |
|---|---|---|---|
| 0.792 | 0.760 | 0.005 | 0.613 |

The low $AP_{95}$ and mIoU implied some omission and commission errors exist in net detection results. Therefore, a classification integrating the results of Faster and Mask R-CNN may be the potential solution. The classification included two steps. First, the Mask R-CNN modelled the area covered with nets, and then the zonal statistics were used to sum up the mapped area in each field. If more than half of the area is within a field covered with nets, this field is temporarily classified as being covered with nets. Second, the zonal results of Mask R-CNN were overlaid with those of Faster R-CNN. If a field with less than 7200 plants/ha detected by Faster R-CNN in Section 3.1 is also classified as covered with nets, this field is finally classified as being covered with nets or other growth stages of pineapples.

After integrating the field-based results of hat and net mapping obtained in Sections 3.1 and 3.2 above, accuracy assessments are presented in Table 6, with the categories defined as "covered with nets," "wearing a hat," and "other stages" that are not in the first two categories. The results revealed that the overall classification accuracy is 97.46%, and the kappa coefficient is 0.908, indicating that the pattern interpretation model has a certain degree of credibility and can accurately interpret and cover the state of the pineapples covered with nets. The total planting area of pineapples in the study

area is 152.387 hectares, and the total planting area of pineapples covered with nets is 14.628 hectares. The netting area successfully judged is 11.921 hectares, and the producer's accuracy is 81.49%. About 2.707 hectares were misclassified as being pineapple of other stages, which is the omission error of the proposed process in Figure 8.

**Table 6.** Accuracy assessment of cultivation field-based mapping of sunscreen mature fruits (Unit: ha).

|  |  | Ground Truth | | | Total Area | User's Accuracy |
|---|---|---|---|---|---|---|
|  |  | **Net** | **Hat** | **Others** |  |  |
| CNN results | Net | 11.921 | 0 | 0.232 | 12.153 | 98.09% |
|  | Hat | 0 | 10.367 | 1.053 | 11.419 | 90.78% |
|  | Others | 2.707 | 0 | 126.108 | 128.815 | 97.90% |
|  | Total area | 14.628 | 10.367 | 127.393 | 152.387 |  |
| Producer's accuracy | | 81.49% | 100.00% | 98.99% | Overall accuracy | 97.38% |
|  | |  |  |  | Kappa | 0.9077 |

As mentioned in Section 3.1, when the number of pineapple plants per field was less than 7200, this study was listed as a re-inspection list, and this part was corrected based on the results of pineapple morphology judgment. The correction targets are shown in Figure 8. If the field is netted, it will be included in the area covered by nets; if the pineapple state is not judged to be netted, it will be included in the area pineapples at other stages. Finally, based on the whole proposed process, this study showed that the area of the field with hats in the study area was 11.419 hectares, and the area covered with nets was 12.153 hectares. The spatial distribution of the above two states is shown in Figure 10. In other words, the total area of sunscreened mature fruits is 23.572 hectares. The feasibility of the yield estimation based on these detection results is discussed in the next Section 3.3.

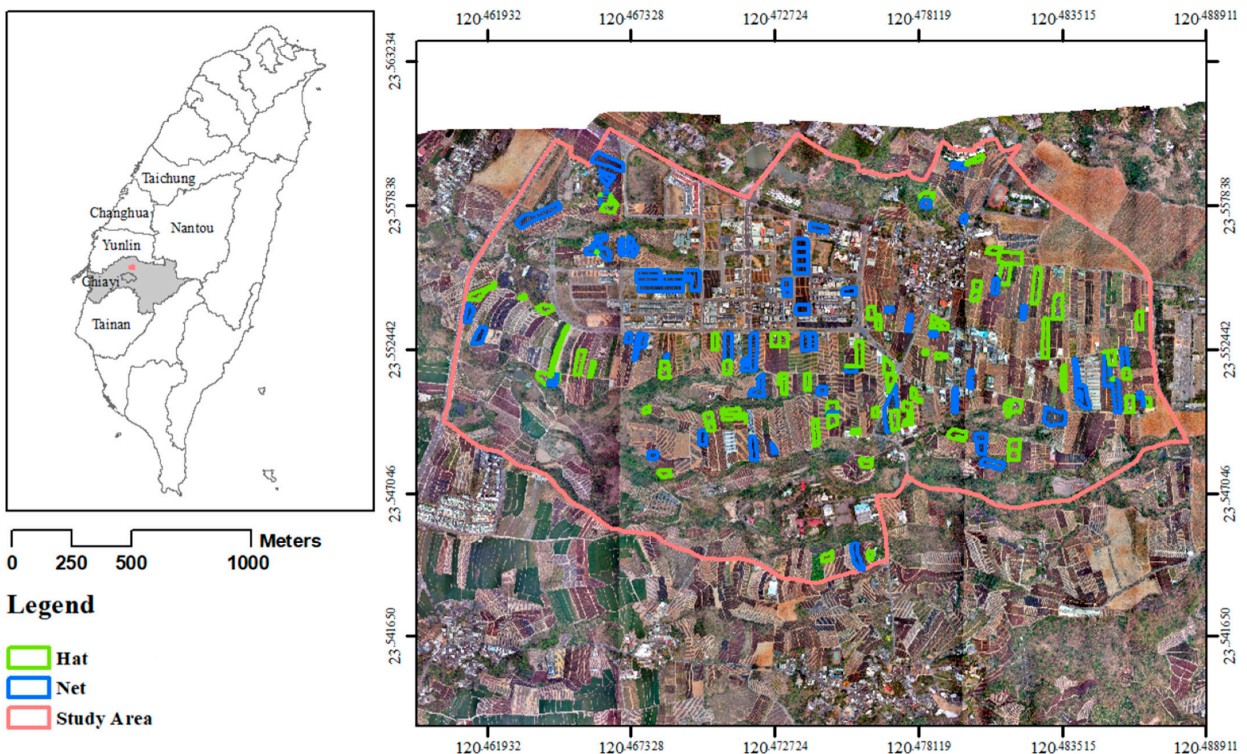

**Figure 10.** The interpretation results of pineapple fields wearing a hat and covered with nets.

### 3.3. Yield Estimation

The F1-score (0.849) with IoU > 0.5 in Section 3.1 implied that some omission and commission error exists in hat wearer detection, and we cannot know the location and quantity of large fruits under the nets. Therefore, it is necessary to understand the number of pineapple plants per unit area in the study area to complete the yield estimation of pineapple in the study area. Therefore, the 35 fields with the highest accuracy of hat wearer detection were selected as the demonstration area. The average number of planted plants per unit area was calculated for the fields with sunscreened mature fruits as the reference for the whole study area. In these 35 fields, the average number of plants per unit area of sunscreened mature fruits was estimated to be 37,467 per hectare by dividing the number of plants calculated by hat wearer detection by the total area of these 35 fields.

Next, if the average weight of each mature fruit at the harvesting stage can be obtained, the total production in the study area can be established. Although the official historical statistics provide the average weight of each fruit in several local fields, fruits are easily affected by the climate after flowering and the farmer's field management efficiency, resulting in problems such as ultra-small fruit, bad fruit, and sunburn in the field. Meanwhile, from January to April, before the shooting of UAV imagery, there was less rain in the study area. Although this was beneficial to the growth of pineapple fruit, it also brought about the problem that the fruit weight was lighter than expected. Therefore, we still picked and weighed the fruit on-site on 4 and 17 May 2021. The same 15 from the 35 fields as mentioned in the previous paragraph were also selected, while we also included another 21 fields for fruit picking and weighing. The other 21 fields were used mainly because not all of the 35 fields were harvested before mid-May, so it is necessary to find other fields that were being harvested. The total number of sampled fields for fruit weighing was 36 (shown as green points in Figure 1), and therefore the yield ranges can be estimated using interval estimation of known population and population variation. Single pineapple weight was estimated as $1.570 \pm 0.080$ kg with a 95% confidence level. The data relating to production estimation are arranged in Table 7.

**Table 7.** The data used for production estimation.

| Data Resource | Item | Data |
|---|---|---|
| Weighing data from field survey and official statistics | Number of samples | 36 |
| | Average weight of one fruit (kg) | 1.570 |
| | Standard deviation of the weight of one fruit (kg) | 0.246 |
| | Weight of one fruit at the 95% confidence level (kg) | $1.570 \pm 0.080$ |
| CNN results | Yield of sunscreen mature fruit (number/ha) | 37,467 |
| | Total area of sunscreen mature fruit (ha) | 23.572 |

Finally, we considered that the yield per hectare was the average number of plants multiplied by the weight of a single pineapple under the 95% confidence level and then multiplied by the 10% loss rate. The yield range per hectare in the study area was about 50,243 to 55,638 kg/ha, which was close to the records from FAOSTAT 2020 and Agricultural Situation Report Resource Network in 2020, mentioned in Section 2.1. The total production range harvested in the short term (about two to three months) was 1,184,327.996 kg to 1,311,498.936 kg.

## 4. Discussion

Comparing the advantages of this study with other studies, there are some references related to fruit detection; nonetheless, few studies have focused on pineapples and UAV imagery. Based on the limited literature and references, similar approaches have applied UAV imagery for analysis; the detection algorithms include object-oriented machine learning classification methods [3] and the deep learning Keras-RetinaNet model [2]. Segmentation is the most critical procedure for object-oriented machine learning classification; in this

procedure, pineapple and non-pineapple plants should be identified and separated as well as possible. Artificial neural network (ANN), support vector machine (SVM), random forest (RF), naive Bayes (NB), decision trees (DT) and k-nearest neighbors (KNN) were then used to classify the UAV images. When dealing with the images shot at a low altitude of 3 m height, a previous study found that the performance could achieve between 85% and 95% in terms of overall accuracy [3]. Concerning the previous study with the deep learning Keras-RetinaNet model, the model and the shooting conditions of UAV images were similar to this study; however, GSD and the accuracy assessment were not presented in detail [2]. To summarize, after a comprehensive review of the above literature, the image spatial resolution shows differences, so accuracy might not be comparable considering different flight. On the other hand, imagery shooting on the ground [48] or UAV with low altitude [3] have high spatial resolution and may contribute to the high accuracy. However, the critical issues of ground shooting are that it is more likely to be blocked by leaves, and the field of view is limited. Some issues do exist within the situation of UAV low-altitude shooting such as small field of view, and the applications might be suitable for small-scale field management.

Nevertheless, compared with the literature [2,3] related to pineapple detection, we can properly estimate the yield of pineapple through computing the number of pineapples per unit area, mapping the cultivated area of the whole region and in situ sample weighing during the early stage of the whole harvest period. If the authorities, i.e., Council of Agriculture and its affiliated agencies, could apply such procedures to estimate the yield of pineapples for an area of hundreds of hectares, this is also the main contribution of this study.

From the above 3 cm-resolution aerial image, it is feasible to detect the sunscreened mature fruits. However, the detection of other growing stages, such as the planting stage, becomes a difficult challenge. There are even some mature fruits covered with nets that are confused with planting stage. This is because the newly planted pineapple plants blended into the background more seriously, and the characteristics of the plants were less clear. This makes it more challenging to judge whether it was an artificial or a computer interpretation process. The super-resolution (SR) method may be helpful for improving the resolution of the images, which not only reduces the cost of shooting but also helps to increase the accuracy of interpretation. High-resolution images contain many detailed textures and critical information. Under the constraints of software, hardware, and cost, SR is considered one of the most effective methods to obtain high-resolution images from single or multiple low-resolution images [49]. The SR process for optical remote sensing imagery includes unsupervised perspective based on the convolutional generator model without the need of external high-resolution training samples [50], supervised perspective based on wavelet transform (WT) and the recursive Res-Net [51] or generative adversarial networks (GAN) [52]. Therefore, using small-scale high-resolution images to estimate the spatial characteristics of pineapple plants from other low-resolution images is the direction for sustainable improvement in the future.

## 5. Conclusions

A large-scale shooting of UAV images of the test target at 3 cm spatial resolution was carried out on 4 April 2021. At the same time, ground investigators were dispatched to the local area during the UAV shooting to record the growth of pineapples to facilitate the reference for subsequent model sample construction. Considering the growth pattern of pineapples affecting the yield, we chose to interpret two categories of sunscreened mature fruit: wearing a hat and covered with nets. Faster R-CNN was first used to detect the plants wearing a hat. Mask R-CNN detected the covered field area and used interval estimation to estimate the yield and production in the study area.

The contribution of this study is that we demonstrated the importance of combining two different CNN methods, object detection and sematic segmentation, as well as zonal statistics and understanding local planting characteristics. If only a single model is used,

the types of features that can be detected are limited. The fruit wearing a hat is suitable for object detection because the shape of the object is consistent and it is easy to distinguish from the background. The fruits covered with nets have a wider range and irregularity, but with a more homogeneous color, and so are suitable for processing with sematic segmentation. After combining the concept of zonal statistics, more commission errors of object detection can be filtered out, and the overall accuracy of sematic segmentation can be improved, thus overcoming the potential issue of farmers' harvesting in different batches.

This study focused on mature fruit counting. At the same time, considering that the less overlapping leaves of seedlings in the planting stage are easier to distinguish from varied plants, fruit counting in the planting stage may also be feasible. Future studies could utilize the monitoring data in the planting stage to further estimate the plant loss rate from the early to mature stage, which would be more likely to estimate the yield in advance.

**Author Contributions:** Conceptualization, R.-Y.L. and Y.-S.S.; methodology, Y.-S.S.; software, Y.-C.C.; validation, Y.-S.S., R.-Y.L. and Y.-C.C.; formal analysis, R.-Y.L.; investigation, Y.-C.C.; writing—original draft preparation, Y.-S.S.; writing—review and editing, R.-Y.L.; visualization, Y.-C.C.; project administration, Y.-S.S.; funding acquisition, R.-Y.L. All authors have read and agreed to the published version of the manuscript.

**Funding:** This research was funded by Agriculture and Food Agency, Council of Agriculture, Executive Yuan, Taiwan, grant number 110AS-3.1.1-FD-Z4. Some funding was partially supported by the National Science and Technology Council, Taiwan (Grant Number: NSTC 111-2410-H-035-020-, and NSTC 111-2119-M-008-006-).

**Data Availability Statement:** The data presented in this study are available on request from Central Region Branch, Agriculture and Food Agency, Council of Agriculture, Executive Yuan, Taiwan.

**Acknowledgments:** The authors gratefully acknowledge the Central Region Branch, Agriculture and Food Agency, Council of Agriculture, Executive Yuan, Taiwan for its assistance in field surveys and farmer interviews. The authors also thank Tzu-Tsen Lai and Chia-Hui Hsu for field investigation and technical support, and Steven Crawford Beatty and Wu I-Hui for English proof-reading.

**Conflicts of Interest:** The authors declare no conflict of interest.

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
