# Peer review of "Pineapples’ Detection and Segmentation Based on Faster and Mask R-CNN in UAV Imagery"

_remotesensing, doi:10.3390/rs15030814_

Round 1
Reviewer 1 Report
This paper proposed Pineapples’ Detection and Segmentation Based on Faster and Mask R-CNN in UAV Imagery. Overall, the structure of this paper is well organized, and the presentation is clear. However, there are still some crucial problems that need to be carefully addressed before a possible publication. More specifically,
1. A deep literature reviews should be given, particularly advanced and SOTA deep learning or AI methods in remote sensing. Therefore, the reviewer suggests discussing some related works by analyzing the following papers in the revised manuscript, e.g., 10.1109/TGRS.2020.3015157.
2. Some important methods should be highlighted.
3. Some future directions should be pointed out in the conclusion.
Reviewer 2 Report
Dear authors,
Your MS is agree with the scope of Remote Sensing journal. Your study is very interesting to help farmers to obtain data to manage their fields, before harvest.
However, you need modify critical aspects in your MS, as Introduction, Objectives, Material&Methods, and Conclusions, mainly. Several comments locate in the attach file, should be respond in detail. Moreover, English require a detailed revision, per native speakers.
Cordial greetings

Reviewer 3 Report
In this paper, a combined Faster R-CNN and Mask R-CNN model was used to detection the Pineapple in UVA images. The authors have put a lot of work into in data acquisition. I’m quite convinced about the practicality of this methodology. However, the authors should further highlight the innovation. Some of the main problems are summarized below:
1. In Introduction: Deep-learning based detection models has been widely used in many field and remote sensing scenarios. Some articles about application of CNNs ( Faster R-CNN, YOLOs) in Remote Sensing should be introduced in line 56 to 72, such as: Application of Combining YOLO Models and 3D GPR Images in Road Detection and Maintenance; Automatic Detection of Pothole Distress in Asphalt Pavement Using Improved Convolutional Neural Networks.
2. Figure 1 and Figure 10. The legend referred to “Red is Band_1, Green is Band_2, and Blue is Band_3”. However, it is difficult for the reader to find these three legends in these figures, and they are not described in the article.
3. The author does not clearly list the number and division of the data set, which confuses the reader.
4. As for Mask R-CNN network, the mIoU should be reported instead of F1 score and AP because it is more reasonable to consider false positive and false negative as the authors described the formula. Besides, this paper should be discussed, which contained the comprehensive evaluation indicators (mIoU, AP50, AP75, AP95, et al.): Automatic pixel-level detection of vertical cracks in asphalt pavement based on GPR investigation and improved mask R-CNN. https://doi.org/10.1016/j.autcon.2022.104689.
5. Please give the network parameters of Faster and Mask R-CNN.
6. Results and discussion are relatively weak, lack comparison with other mainstream models used for pineapple detection?
7. In Abstract, the author said “the F1 score of the hat wore detection results was 0.849, the AP was 0.739, the precision 16 was 0.990, and the recall was 0.743”. However, we did not see the F1 these results in the MS. Please check it. You’d better list the result in a table.
8. What is the detection speed of the proposed model?
9. Please revise language editing of the manuscript, some errors were found.
Round 2
Reviewer 1 Report
No more comments.
Reviewer 2 Report
Dear authors,
The latest version of the work has collected a good part of the comments and suggestions made, so the work has improved.
Although, there is a clear deficiency in the discussion section. The authors reintroduce the results (lines 564-599), which is not expected in this section. The content exposed in lines 600 to 632, includes 6 references to previous works, where no comparison is made since the spatial resolution is different in these studies. Any comment on the results, or the search for other references is highly recommended.
In line 612, it mentions 'compared with other literatures...', however it does not state which ones they refer to.
Line 601 - Review text: ...few studies dwell on...
It is therefore recommended to rewrite the discussion section.
Cordial greetings.
Reviewer 3 Report
Summary
The author has done a good job with the revisions, I have a few remaining further questions/remarks before publication.
1. In abstract, the mIoU with input image size should be described.
2. In Introduction: Deep learning based detection models has been widely used in many field and r emote sensing s cenario s. Some articles about application of CNN s Faster R CNN, YOLOs) in Remote Sensing should be introduced in line 56 to 72, such as:
(1) Automatic recognition of pav ement cracks from combined GPR B scan and C scan images using multiscale feature fusion deep neural networks
Automation in Const ruction . DOI: 10.1016/j.autcon.2022.104698
(3) Ultra High Resolution UAV Based Detection of Alternaria solani Infections
in Potato Fields . Remote Sensing . DOI 10.3390/rs14246232
3. All figures should provide as clear a vector image as possible
